# Cats and SARS-CoV-2: A Scoping Review

**DOI:** 10.3390/ani12111413

**Published:** 2022-05-30

**Authors:** Ramona Doliff, Pim Martens

**Affiliations:** University College Venlo, Maastricht University, Nassaustraat 36, 5911 BV Venlo, The Netherlands; m.doliff@student.maastrichtuniversity.nl

**Keywords:** SARS-CoV-2, COVID-19, cats, systematic literature review, scoping review

## Abstract

**Simple Summary:**

Cats are susceptible to SARS-CoV-2. Considering the close contact that exists between humans and cats, this is worrisome; virus transmission between species carries a particular risk of leading to the development of new virus variants. Therefore, the aim of this review was to provide a comprehensive overview of what is known concerning the role of cats in the spread of SARS-CoV-2, to guide further research and inform policymakers. The main outcome of this review was that, while cats are susceptible to the virus, and transmission from humans to cats happens regularly, there is currently no evidence of widespread SARS-CoV-2 circulation among cats. Overall, cats seem to play a small role in the spread of the virus. Nevertheless, this review also revealed substantial gaps in research. For instance, large-scale studies including more cats are needed to solidify evidence gathered from individual studies. Moreover, the role of stray, feral, and shelter cats has attracted little research, as well as the possibility of cat-to-cat virus transmission beyond experimental infection. Tackling these gaps in the research is important to adequately evaluate the danger of cats’ susceptibility to SARS-CoV-2, now and in the future.

**Abstract:**

Since the beginning of the COVID-19 pandemic, various animal species were found to be susceptible to SARS-CoV-2 infection. The close contact that exists between humans and cats warrants special attention to the role of this species. Therefore, a scoping review was performed to obtain a comprehensive overview of the existing literature, and to map key concepts, types of research, and possible gaps in the research. A systematic search of the databases PubMed, Google Scholar, and Scopus and the preprint servers medRxiv and bioRxiv was performed. After a two-step screening process, 27 peer-reviewed articles, 8 scientific communication items, and 2 unpublished pre-prints were included. The main themes discussed were susceptibility to SARS-CoV-2, induced immunity, prevalence of infection, manifestation of infection, interspecies transmission between humans and cats, and lastly, intraspecies transmission between cats. The main gaps in the research identified were a lack of large-scale studies, underrepresentation of stray, feral, and shelter cat populations, lack of investigation into cat-to-cat transmissions under non-experimental conditions, and the relation of cats to other animal species regarding SARS-CoV-2. Overall, cats seemingly play a limited role in the spread of SARS-CoV-2. While cats are susceptible to the virus and reverse zoonotic transmission from humans to cats happens regularly, there is currently no evidence of SARS-CoV-2 circulation among cats.

## 1. Introduction

The first cases of coronavirus disease 2019 (COVID-2019) in humans were reported in December 2019 [1]. The causative virus, severe acute respiratory syndrome coronavirus 2 (SARS-CoV-2), spread around the globe rapidly, leading to a worldwide pandemic that now poses one of the defining global health challenges of our time. To date, the zoonotic origin of SARS-CoV-2 has not been conclusively identified. The most highly related coronaviruses were found in bats and pangolins, yet none of the viruses identified are similar enough to SARS-CoV-2 to be its direct progenitor [2]. However, while the animal origin of SARS-CoV-2 remains elusive, the emergence of this novel virus highlights the centrality of human–animal interaction and relationships for the emergence of novel infectious diseases.

Human–animal interactions are not only relevant for the emergence of SARS-CoV-2, but also for the dynamic and future course of the ongoing pandemic [3]. Various animal species have been found to be susceptible to infection with SARS-CoV-2 under experimental conditions, and sporadic cases of natural infection have been reported in different species, including farmed minks, lions, tigers, and cats, among others [4,5]. The susceptibility of a species to this virus is predominantly determined by the structure of the angiotensin-converting enzyme 2 (ACE2) of the respective species [6,7]. Through binding to the spike protein of SARS-CoV-2, the ACE2 protein functions as the main receptor mediating virus entry into the cells of the body.

Transmission of the virus from humans to animals bears the risk of the establishment of SARS-CoV-2 in the animal population [8]. This would enable the further adaption and evolution of the virus in the new hosts, and possibly, the transmission of new virus variants to humans [4,8]. In the case of farmed minks in Denmark, the transmission of a novel strain from minks back to humans has been reported [4,9]. The constant evolution of SARS-CoV-2 and consequent emergence of ever-new virus strains is an unavoidable reality, even if only transmitted among humans. However, cross-species transmission bears a particularly high danger of leading to the emergence of more virulent, transmissible, or pathogenic strains. Furthermore, strains that are more successful at evading host immunity in humans induced by vaccinations or previous infection are more likely to emerge due to adaption to host differences [8]. Thus, it is crucial to take into account the relevance of human–animal interactions for a successful global response to the COVID-19 pandemic.

Several scholars, among them the WHO mission team on the origins of SARS-CoV-2, brought to attention the specific danger of cats as potential SARS-CoV-2 hosts due to their high susceptibility to the virus [2,4,10,11]. The feline ACE2 is one of the most closely related to human ACE2, and the receptor-binding interface region is particularly homologous [7,10]. In addition to cats’ biological susceptibility to SARS-CoV-2, the close contact that exists between many humans and their domestic cats warrants special attention to these animals. Moreover, the fact that many domestic cats roam around freely, thus possibly having contact with different animals, makes it even more important to investigate the role that cats play with regard to SARS-CoV-2. Domestic cats mostly live in their owners’ houses, often in close physical contact with humans, and might have contact with other animals when going out in gardens and beyond. They might have contact with other cats or other wild and domesticated species susceptible to SARS-CoV-2, such as wild feline species, black voles, ferrets, minks, deer, or hamsters [5,12].

There has been considerable research regarding the susceptibility of cats to SARS-CoV-2, specifically under laboratory conditions, considering the short timeframe since the emergence of the virus [12,13,14,15]. However, there has been no thorough evaluation of the role of cats with regard to the spread of SARS-CoV-2 and the risk of the establishment of SARS-CoV-2 in cat populations in light of the most recent literature.

The COVID-19 pandemic is one of the defining global health challenges of our time. To understand the future spread, circulation, and development of SARS-CoV-2, beyond the acute pandemic phase, it is vital to look at the virus’ impact beyond the human species. Cats as potential hosts for SARS-CoV-2 are present in and around many households globally. Thus, the evaluation of the risks that interactions between humans and animals pose to cats themselves, humans, and other animals regarding the spread and evolution of SARS-CoV-2 is of paramount importance. The potential role of these companion animals for SARS-CoV-2 dynamics is a subject of urgent concern. Scoping, synthesizing, and furthering the existing evidence are important to inform further research, but also to aid the decision of policymakers. The research question in this scoping review is: What is the role of cats with regard to the spread of SARS-CoV-2?

## 2. Methods

### 2.1. Data Sources and Search Strategy

Initially, explorative searches in Google (Scholar) and PubMed were performed to define the search terms. The search terms were divided into two groups relating to cats (i.e., cats, cat) and SARS-CoV-2 (i.e., SARS-CoV-2, COVID-19). Within each group, the search terms were linked using the Boolean operator “OR”, and both groups were combined using the operator “AND”.

On 15 September 2021, the electronic databases PubMed, Google Scholar, and Scopus were searched for publications containing the search terms in the title and/or abstract. In addition to peer-reviewed articles, the initial explorative search revealed potentially useful records on preprint servers, which led to the decision to include the preprint servers medRxiv and bioRxiv as additional databases in the search. The same search terms were used for the identification of relevant preprints as peer-reviewed publications. The exact search strings used for each database can be found in Appendix A.

### 2.2. Study Selection

All citations identified in the search were imported into the citation manager EndNote X9 (Clarivate Analytics, London, UK). Citations were then imported into the web-based systematic review software Rayyan (Rayyan Systems Inc., Cambridge, MA, USA). Duplicate citations were removed manually. Subsequently, a two-stage screening process was performed to assess the relevance of sources identified in the initial search. First, a title and abstract relevance screening was performed. Second, the selected articles were reviewed in the full text.

To assess the relevance of the studies for this review, inclusion and exclusion criteria were defined, a full list of which can be seen in Table 1. All study designs were included in this review. However, only primary research articles were included, meaning the exclusion of secondary research such as review articles, commentaries, and editorials. The topic area of articles to qualify for inclusion, broadly speaking, was the role of cats and their interaction with humans and other animals with regard to the spread of SARS-CoV-2. However, a variety of topic areas were excluded for different reasons. First, articles that dealt with the assessment of cats as animal models for human SARS-CoV-2 infection were excluded, as this review is interested in the role of cats, not their use to mirror human SARS-CoV-2 infection. Second, articles primarily dealing with human well-being derived from contact with their pet cats were excluded, as, again, the focus of this review is the role of cats, not human wellbeing. Third, articles concerning the development or evaluation of biochemical assays for SARS-CoV-2 for cats or using cats, such as diagnostic PCR assays or ELISA assays, were excluded, as these are biochemical topics that are beyond the scope of this article. For the same reason, in vitro and in silico biochemical analyses relating to the cat’s ACE2 receptor, such as structural analysis or binding affinity assays, were excluded. Furthermore, articles that did not singularly focus on cats and their interaction with humans or other animals, but that included multiple animal species as study participants, were excluded. Importantly, articles that dealt with the relation of cats to other animals were specifically not excluded, but only articles that included multiple animal species next to each other, so for instance, studies that had both dogs and cats as study participants. The main reason for this decision was the feasibility of the review. As there were many studies that included a variety of animals next to each other, the inclusion of all these articles, while certainly informative, would not have been viable in the context of this review.

### 2.3. Data Characterization, Summary, and Synthesis

To extract key items of information of each selected article, a data charting form was developed by the authors using Microsoft Excel 2010 (Microsoft Corporation, Redmond, WA, USA). Using the extracted information, the results were presented in a narrative form. The results were collated and summarized in terms of the different relations that were researched. First, the results regarding the relation between cats and SARS-CoV-2 were collated under different aspects, namely the susceptibility of cats to SARS-CoV-2, SARS-CoV-2-induced immunity, and disease pathology. Second, the results regarding the relation of humans and cats with regard to the virus were collated, and third, the results concerning the relation between cats with regard to SARS-CoV-2.

## 3. Results

### 3.1. Article Selection

Combining all four databases, the search strategy yielded 761 records in total, including both peer-reviewed articles and preprints. After duplicate removal, 404 records were title–abstract screened. Moreover, 43 full-text records were then assessed for eligibility, of which 36 records were included in the final scoping review. The article selection process can be seen in more detail in Figure 1 [16].

### 3.2. General Information about the Included Reports

Of the 36 records included in the final review, 75% (27/36) were peer-reviewed articles and 5.6% (2/36) were unpublished pre-prints. Additionally, 19.4% (8/36) were different scientific communication items published in scientific journals, including letters to the editor, research letters, and rapid communications. The peer-reviewed articles and scientific communications were published in 18 different scientific journals. The studies took place in 16 different countries in total, which can be seen in Figure 2. However, more than half of the studies took place in only four countries: Italy, China, Germany, and the USA. Moreover, 25% of the reports (9/36) were published in 2020, and the remaining 75% (27/36) in 2021. Regarding the type of study, there were 20 case studies (55.6%), 10 cross-sectional studies (27.8%), and 6 experimental studies (16.7%). No case-control or cohort studies were among the eligible reports. The 36 pieces, including studies, their aims, and study types can be seen in Table 2.

### 3.3. Cats and SARS-CoV-2

#### 3.3.1. Susceptibility of Cats to SARS-CoV-2

The first relation frequently explored within the articles, was the relation between the virus and cats. The most prevalent topic was the susceptibility of cats to SARS-CoV-2. Four different experiments showed that cats are highly susceptible to SARS-CoV-2 infection when inoculated intranasally or orally, and show prolonged periods of viral shedding [14,17,20,23,45].

Not only has the susceptibility been shown in experimental settings, but many infections of cats that occurred naturally have been recorded; Numerous studies included in this review reported the human-to-cat transmission of SARS-CoV-2, a topic which will be explored further in Section 3.4. However, not in all studies of natural infections, was the source clearly human. Both Musso et al. [31] and Brandão et al. [19] reported SARS-CoV-2 infection in a domestic cat with an unknown source of infection, in Italy and Brazil respectively. While there was no clear evidence that these infections are from a human source, there also was no evidence contrary to this.

Additionally, three case studies detected the alpha variant of SARS-CoV-2 in each case of natural infection, confirming the susceptibility of cats to this variant [22,26,42]. None of the included studies reported the detection of any other variant of concern.

#### 3.3.2. SARS-CoV-2-Induced Immunity and Seroprevalence among Cats

Not only did various studies evidence the susceptibility of cats to SARS-CoV-2 infection, but they also evidenced antibody development against SARS-CoV-2 after confirmed infections. Under experimental conditions, all cats seroconverted and developed antibodies [14,23,24]. However, under natural conditions, different case studies showed that, while most cats developed antibodies, some did not [18,21,27,28,32,33,35,43,47,48,50]. Specifically, Bessière et al. [18], Chaintoutis et al. [21], Neira et al. [33], and Klaus et al. [28] reported detectable antibody development in only one out of two investigated cases per study.

The duration of seropositivity was tested in a few of the included studies with heterogenous results. For instance, in a case study by Klaus et al. [27], SARS-CoV-2 antibodies were detectable for six months. Yet, in the case study by Pagani et al. [34], there were no antibodies detected in a follow-up appointment six months after the initial infection. Zhang et al. [41] regularly sampled the serum of one cat found positive for SARS-CoV-2 antibodies in a serosurvey in China. The antibody levels declined below the limit of detection after 110 days. However, the time of infection for this cat was unknown.

Among the included studies, there were a number of serological surveys investigating the prevalence of SARS-CoV-2 antibodies in cats. For instance, Mitchelitsch et al. [29] found a prevalence of 0.7% for SARS-CoV-2 antibodies when sampling the sera of pet cats taken by veterinarians during clinical examination all over Germany between April and September 2020. A second study with the same study design was conducted by Mitchelitsch et al. [30] with samples taken at a later stage of the pandemic, between September 2020 and February 2021. This study indicated an approximate doubling of the seroprevalence to 1.4% compared to the first study, while human cases increased eightfold in the same period. The study with the highest seroprevalence of 31.7% was performed by Jara et al. [25]. However, this study only included cats whose owners previously had COVID-19, thus explaining the substantially higher prevalence. Moreover, Zhang et al. [41], who reported a seroprevalence of 10.8%, also included pet cats of owners with a history of COVID-19, among other things. The main results of all serological surveys regarding SARS-CoV-2 antibodies in cats included in this review can be seen in Table 3. Overall, the serosurveys are hard to compare for different reasons, as elaborated in the discussion. Yet, despite a lot of uncertainty and differences between the studies, the serological surveys indicate a relatively low prevalence of SARS-CoV-2 antibodies in cats overall.

Two experimental studies tested whether the induced immune protection, including antibody production, provided by previous infection, protects from reinfection. Chiba et al. [45] detected no virus replication in respiratory organs and no additional lung damage in their reinfection study. Gaudreault et al. [24], on the other hand, found that reinfection was possible as transient viral shedding was observed. Yet, the shedding periods were consistently shorter than when first infected, and viral DNA was detected from fewer tissues and at lower levels when animals were dissected. Moreover, the reinfected cats did not transmit the virus to any of the co-housed cats. Thus, SARS-CoV-2 infection appeared to provide at least a partial, yet non-sterilizing immune protection against reinfection with the virus.

#### 3.3.3. Manifestation of Infection

One common theme in the included studies was the clinical manifestation of SARS-CoV-2 infection in cats. Specifically, many case studies reported the clinical symptoms in infected cats. Many of them report respiratory symptoms. Brandão et al. [19], Fritz et al. [50], and Gargliany et al. [47] generically reported respiratory symptoms, while Neira et al. [33], Pagani et al. [34], and Zoccola et al. [42] reported mild respiratory symptoms. Other studies more specifically described sneezing, coughing, and/or shortness of breath [27,28,38,44]. Some cats developed symptoms of pneumonia. Hosie et al. [11] reported the case of a kitten that was euthanized due to severe respiratory symptoms resulting from SARS-CoV2-induced pneumonia. Moreover, Natale et al. [32], Keller et al. [26], and Musso et al. [31] described cases with symptoms and signs of pneumonia. Notably, in contrast to Hosie et al. [11], they did not perform a histological analysis to conclusively confirm SARS-CoV-2 as a causal agent. The cat Musso et al. [31] reported on eventually died due to severe clinical manifestations.

Non-respiratory symptoms that were reported in different case studies included:Gastrointestinal symptoms [32,38,47],cardiac abnormalities [22],inappetence and apathy [28,38],ocular discharge [27,38], andfever [26,38].

Three case studies reported no clinical signs in infected cats at all [18,21,33]. However, one must keep in mind that testing for SARS-CoV-2 in many case studies was performed in response to symptoms in cats, thus likely underestimating the number of subclinical infections. Regarding experimental studies, Bao et al. [17] reported arching of the back and diarrhea in some cats, while three other experimental studies did not find any clinical signs [14,23,45]. All taken together, the studies show a relatively heterogenous picture of the clinical manifestations in SARS-CoV-2-infected cats. Yet, most studies reported, if any, only mild clinical manifestations in SARS-CoV-2-infected cats, and the most prevalently reported signs were respiratory symptoms.

Some studies provide evidence that comorbidities play a role in the manifestation of clinical disease. For instance, the case study by Klaus et al. [27] describes a cat with lymphoma that develops severe respiratory symptoms, including sneezing and coughing, as well as ocular discharge after SARS-CoV-2 infection. Moreover, Villanueva-Saz et al. [40] suggest that immunosuppressed animals might be especially susceptible to SARS-CoV-2 infection as three out of four seropositive cats have concomitant infections with other pathogens.

The duration of infection and viral shedding profiles varied greatly between different cats and different studies. Regarding shedding profile, most studies reported higher viral loads and longer viral shedding in oropharyngeal or nasal swab compared to lower (or no) detectable viral loads in rectal swabs [14,18,22,28]. Only one case study reported a cat being SARS-CoV-2 positive only in the rectal swab and not in the oropharyngeal swab [44]. Curukoglu et al. [22] also took eyelid swaps which were all negative. One case study found fur and bedding swabs to be SARS-CoV-2 positive in addition to oropharyngeal and rectal swabs [28]. Considering the cumulated evidence, oropharyngeal and nasal swabs appeared to be the most relevant samples to reliably determine SARS-CoV-2 infection in cats using polymerase chain reaction (PCR) techniques.

Regarding duration of viral shedding, studies show heterogenous results. Bessière et al. [18] for instance only report weak and transient shedding in two cases of natural infection. Other studies of natural infections report more prolonged periods of shedding; Neira et al. [33] report 5–17 days, Chaintoutis et al. [21] 7 days, Natale et al. [32] approximately 2 weeks and both Schulz et al. [48] and Zoccola et al. [42] report 3 weeks of viral shedding. Regarding experimental studies, Halfmann et al. [14] report 4–5 days of shedding, Gaudreault et al. [23] 8–10 days, and Bao et al. [17] measured at least 14 days. Overall, shedding periods seem to differ substantially between cases. However, one also must consider that the mean of detection of the viral load differed between the different studies, thus making the exact numbers only limitedly comparable.

One thing that is important to note is that SARS-CoV-2 was often detectable before onset of clinical signs, as shown for instance by Klaus et al. [27]. The duration of clinical symptoms reported in the different studies varied greatly from a few days to a few weeks. The longest duration of symptoms was recorded by Natale et al. [32], who reported 2 weeks of symptoms in one case and Subotsina et al. Subotsina, Gromov and Kupryianav [38], who found an average symptom duration of 2–3 weeks in 15 examined cases.

Different experimental studies investigated the pathoanatomical and histological changes in the body of SARS-CoV-2-infected cats. Gaudreault et al. [23] observed pathological changes in the upper and lower airways in experimentally infected cats, specifically mild-to-moderate neutrophilic lymphocytic tracheobronchoadenitis in association with SARS-CoV-2 RNA and antigens. Similarly, Bao et al. [17] found mild-to-moderate bronchiolitis in infected cats as well as expression of SARS-CoV-2 in the intestines. Chiba et al. [45] also found mild bronchiolitis in infected cats and while infectious virus cleared from lungs within six days after initial infection, lung lesions persisted at least until a month after infection. Next to these experimental studies, there was also a big case series investigating the clinical and pathomorphological picture in 15 symptomatic cats with natural SARS-CoV-2 infection in Belarus [38]. The main pathoanatomical changes found here were membranogenic pulmonary edema, pronounced blood clotting in medium-sized blood vessels, and the expansion of the heart chambers and pulmonary vein system, while the main histological features were lymphoid-macrophage peribronchitis and perivasculitis.

### 3.4. Humans to Cat Transmission

A second relation commonly researched was the relation between humans and cats with regard to SARS-CoV-2. Most of the studies included in this review were case studies that reported different cases of the natural human-to-cat transmission of SARS-CoV-2 [11,18,21,22,27,28,33,34,35,38,42,44,48,50]. In many of these studies, the hypothesis of owner-to-cat transmission was underlined through sequencing of the viral genome from both the owner and cat. All sequence comparisons revealed identical or slightly differing genome sequences between owners and their cats [18,21,33,34,35,42,44,48].

These case studies evidenced that human-to-cat transmission happens sporadically, given the close contact between humans and cats. Two cross-sectional studies concerning households with a history of COVID-19 that own a cat further hint at how frequently these transmission events might happen. Barrs et al. [43] performed PCR testing for SARS-CoV-2 on 50 domestic cats of owners with an ongoing SARS-CoV-2 infection in Hong Kong, China. Of all the tested cats, 12% (6/50) tested positive for the virus. Jara et al. [25] found a seroprevalence of 31.7% among cats from households with a history of COVID-19 infection.

Overall, the included studies provided plenty of evidence for the human-to-cat transmission of SARS-CoV-2. Yet, none of these studies found any evidence for the cat-to-human transmission of SARS-CoV-2, indicating that cat-to-human transmission of the virus is highly unlikely. Thus, looking at the relation between humans and cats with regard to SARS-CoV-2, the virus can be seen as an agent traveling in only one direction, from human to cat.

### 3.5. Cat to Cat Transmission

A third relationship that was explored by a few studies is the relation of cats among themselves with regard to SARS-CoV-2. However, there are considerably fewer studies concerning this topic compared to the other two relationships that were explored.

Among the included studies, there were three different experimental studies that investigated the potential of cat-to-cat transmission of SARS-CoV-2. These studies showed that under experimental conditions, cats could transmit the virus to other cats when they were in close contact for prolonged times [14,17,23]. However, in the serial passage transmission experiment over five generations performed by Bao et al. [17], transmissibility and pathogenicity decreased with ongoing passaging. Another important finding of these experimental studies was that asymptomatic cats were capable of transmitting the virus [14,17,24].

One study also provided evidence for a case of cat-to-cat transmission that occurred naturally in a household setting among two domestic cats [19]. A domestic cat returned back to its owner after missing for two weeks, exhibiting respiratory symptoms. A RT-qPCR test of a nasal swab revealed the SARS-CoV-2 infection of the cat, while the owner was SARS-CoV-2 negative and could be excluded as source of infection. The second cat living in the same household was SARS-CoV-2 negative at first, but tested positive at a later point, while reportedly not having been in contact with other animals or humans besides the owner.

Another study by Akhmetzanov et al. [49] that investigated the possibility of natural cat-to-cat transmission of SARS-CoV-2 designed a model for possible sustained chains of transmission among domestic and stray cats in Wuhan based on serological data by Zhang et al. [41]. The study estimated a basic reproduction number R_0_ of SARS-CoV-2 among cats of 1.09 (95% CI: 1.05, 1.13). This means that every infected cat infected 1.09 other cats on average. Akhmetzhanov et al. [49] stated that it was unlikely that sustained transmission occurred among cats in Wuhan, as the R_0_ estimate for cats was low at 1.09 compared to R_0_ = 2.4 for humans in the same area at the same time. Based on this estimate the probability of a major outbreak was calculated to be 7.9%. However, there were some major concerns regarding the methodology of this unpublished preprint that are discussed later.

One experimental study by Braun et al. [20] was concerned with the evolution of SARS-CoV-2 in cats. The researchers performed the experimental modeling of the evolutionary capacity of SARS-CoV-2 between cats and within cats based on a transmission experiment with cats and the regular genome sequencing of the virus present in the different cats. Overall, the study showed that stochastic processes like narrow transmission bottlenecks and genetic drift are the major forces that shape viral genetic diversity within and between hosts, constraining the overall pace of virus evolution [20]. One interesting observation was that a notable variant at amino acid position 665 in Spike (H655Y) arose rapidly and consistently and was transmitted among cats during the experiment, suggesting the positive selection of this variant in cats [20].

## 4. Discussion

### 4.1. Key Findings

The research question that this scoping review sought to answer was: What is the role of cats with regard to the spread of SARS-CoV-2? The reviewed literature provided a wide body of evidence answering this research question to some extent. However, there were also clear gaps in the research. Overall, cats seem to play a limited role in the spread of SARS-CoV-2. They are certainly susceptible to the virus and there is the continuous occurrence of reverse zoonotic transmission from humans to cats, but the studies provided no evidence of SARS-CoV-2 circulation among cats or from cats to humans. In the following, the main themes, types of evidence available, and gaps in the research that emerged from the literature review are summarized and discussed in more detail.

#### 4.1.1. Main Themes

*Susceptibility*, *induced immunity*, and *prevalence*. One of the main themes that emerged from the literature review was the susceptibility of cats to SARS-CoV-2. Overall, the included studies showed that cats are susceptible to SARS-CoV-2 in experimental and non-experimental settings. Two other related themes are the induced immune response by SARS-CoV-2 and based on this, the measurement of prevalence of (previous) infection with SARS-CoV-2 among cats. The antibody development in response to SARS-CoV-2 infection has been recorded in many cases, yet duration of seropositivity was not very well documented. Induced immunity by infection has been shown to provide cats with some protection upon re-exposure to the virus yet does not necessarily provide sterilizing immunity. Serosurveys report a relatively low seroprevalence among cats when compared to the human population.

*Manifestation of infection.* Another prevalent theme was the manifestation of infection in cats that was first and foremost explored in cases studies. In summary, cats often do not develop clinical disease, or if so, only show mild clinical symptoms mostly of respiratory nature. Nevertheless, there have been records of severe clinical disease. When infected, cats mostly show prolonged viral shedding in the upper respiratory organs, yet infection duration and viral shedding profiles vary greatly among cats and studies. Pathomorphological investigations confirmed that the main system impacted by the virus was the respiratory tract.

*Interspecies transmission.* Another theme often discussed in the included literature was cross-species transmission of the virus. Anthropogenic transmission from humans to cats by close contact was the only reliably identified source of SARS-CoV-2 infection in cats in the included studies. The studies provided a solid body of evidence for sporadically occurring infection in cats due to close contact with infected owners or other humans. However, there was no evidence of cat-to-human transmission identified.

*Intraspecies transmission.* Lastly, intraspecies transmission of the virus among cats is a theme present in some of the included studies. Cat-to-cat transmission of SARS-CoV-2 has been shown to be possible under experimental conditions. Additionally, one of the included case studies reported a naturally occurring case of intraspecies transmission [17].

#### 4.1.2. Types of Evidence

Independent of the topic of a specific study, scientific evidence comes in a variety of forms and this scoping review included different kinds of scientific evidence. On the one hand, different study types, producing different kinds of scientific evidence, were included. On the other hand, different publication types were included.

*Study type.* Over half of the studies incorporated in this review were case studies studying one or a few cases of SARS-CoV-2-infected cats. While these case studies provide considerable insights, the strength of the evidence is rather low. Due to studying one particular cat or a small series of cases, case reports lack the ability to generalize their findings. Nevertheless, case studies provide important starting points for further research. The strength of many of the included case studies is that they provide detailed descriptions and insights into the course and manifestation of SARS-CoV-2 infection in cats. Comparing and combining the findings of a variety of case studies was valuable to glean information and further insights. In addition to the case studies, ten cross-sectional studies performed PCR testing or serological testing for SARS-CoV-2 antibodies in different study populations. These studies have the advantage of being more generalizable with regard to the cat population in question. Besides these observational, descriptive studies, six experimental studies were included. These experimental studies were specifically useful to explore themes such as cat-to-cat transmission or disease pathology in a controlled environment, providing a high level of scientific evidence. Yet, one must take into consideration that experimental conditions reflect reality only to a certain degree. For instance, cohousing conditions used in experimental settings are not necessarily reflecting real-life situations. Moreover, the inoculation of cats with viral material is often done with considerably higher viral loads than would mimic natural infections, not reliably representing natural transmission events. Inoculation with a high viral load cannot only lead to more and longer-lasting viral shedding in the animal, but also leads to more pathological signs, such as lung lesions, not representative of natural infections. Moreover, it could potentially lead to more prevalent cat-to-cat transmission events of SARS-CoV-2 than in natural scenarios. Additionally, the size of the viral load used was different among the different studies, making them not only less comparable to natural infection situations, but also less comparable to each other. Gaudreault et al. [23] for instance used an unusually high dose and additionally inoculated the animals via multiple routes to securely reach primary infection.

*Publication type.* As not only peer-reviewed articles were included, it was essential to monitor where information was gleaned from. Among the included studies, there were 27 articles published in scientific journals that had undergone peer review. However, the eight articles published as scientific communication pieces in journals underwent a less rigorous review process and the two included preprints by Akhmetzhanov, Linton and Nishiura [49] and Fritz et al. [50] not undergo any kind of formal review. The publication type for each study can be seen in Table 2. While pre-prints can provide important and timely information, the omission of a peer-review process made it all the more important to treat the information gleaned from these articles with caution and appraise the quality of these studies carefully. The scientific communication articles and preprints were appraised cautiously and despite minor concerns, all but one did not elicit major concerns regarding methodology or reporting. However, the unpublished preprint by Akhmetzhanov et al. [49] showed major flaws in methodology; Two underlying assumptions of the model used to calculate R_0_ among cats were homogenous mixing and random independent sampling, which are violated as the data collected by Zhang et al. [41] did not fulfill these criteria at all. Moreover, the study did not account for the fact that there were multiple cases of known or suspected cases of human-to-cat transmission, but only accounts for cat-to-cat transmission in the model [49]. Overall, the data collected by Zhang et al. [41] were unfit for the purpose they were used for by Akhmetzhanov et al. [49].

#### 4.1.3. Identified Gaps in Research

*Large-scale studies.* While the included articles cover a lot of topics, there are also still gaps in research. More studies that include a greater number of cats as study participants are needed as the largest body of literature identified were case studies, reporting only the case of a single cat or of a few cats. Moreover, studies with a longitudinal design would be of use to investigate characteristics such as the duration of antibody-mediated immunity among cats which is one topic that attracted little research until now.

*Underrepresentation of stray*, *feral*, and *shelter cats.* However, there are not only concerns regarding the study type and number of study participants in the included studies. Another concern that emerged after reviewing the included articles regards the type of cats studied; The studies widely focused on pet cats, possibly underrepresenting the role feral and stray cats play. There were only three studies including feral and stray cats, all of which were cross-sectional studies that investigated the prevalence of SARS-CoV-2 infection in stray cats. Two studies by Spada et al. [36] and Stranieri et al. [37] took place in Northern Italy, while Villanueva-Saz et al. [40] investigated the seroprevalence among stray cats in Zaragoza, Spain. An important difference when it comes to the possible circulation of SARS-CoV-2 among cats between domestic to stray and feral cats is that stray and feral cats often live in colonies. Thus, they get more into contact with other cats, providing potentially more opportunities for intraspecies transmission of the virus compared to domestic cats. Besides domestic, stray, and feral cats, there is also a considerable number of cats that is housed in animal shelters. Only one study by van der Leij et al. [39] focused exclusively on shelter cats and Zhang et al. [41] included 46 shelter cats among a total of 102 cats studied. Both studies aimed at measuring the prevalence of SARS-CoV-2 infection among cats in the Netherlands and Wuhan, China respectively. Shelters provide a special situation; While there is a limited amount of cats within a shelter, they often have very close contact with each other for prolonged amounts of time, more so even than stray or feral cats usually are. However, van der Leij et al. [39] only found a seroprevalence of 0.8% among shelter cats in the Netherlands, suggesting no wide-spread circulation of the virus among shelter cats. Overall, the role that stray, feral, and shelter cats play within the circulation of SARS-CoV-2 might be very different to that of domestic cats, warranting more extensive research into these cat populations.

*Natural cat-to-cat transmissions.* Another gap in research is that cat-to-cat transmissions were nearly exclusively researched under experimental conditions. Among the included studies were some studies that investigated SARS-CoV-2 among cats in an experimental setting [14,17,20,23,24,45]. However, for different reasons discussed in Section 4.1.2 these experimental studies are only partially representative for reality. There was only one case study reporting the case of natural cat-to-cat transmission [19]. Certainly, it is possible that there are only a little number of cat-to-cat transmission happening under natural conditions, which is also supported by the serosurveys that found a relatively low prevalence of SARS-CoV-2 antibodies [36,37,40]. Yet, the fact that few studies have investigated natural cat-to-cat transmission, highlights the importance of further studies with appropriate methodologies to further evidence this hypothesis.

*The relation of cats to other animal species with regard to SARS-CoV-2.* One last important gap in knowledge that was identified in this scoping review was that there were no studies investigating the relation of cats with other species than humans or cats themselves. Cats roam in our gardens, cities, and beyond. Next to other cats, there are many different species that are susceptible to SARS-CoV-2 that cats might have contact, when roaming around freely as stray and feral cats, but also many domestic cats do. Ferrets, minks, racoon dogs, black voles, and white-tailed deer are among those susceptible animals [5,12,51]. While literature is indeed scarce on this topic, the fact that this review defined inclusion of other animals than cats as study participants as exclusion criterium might also be partly responsible for the omission of this topic.

### 4.2. Positioning This Review within the Current Academic Debate

Overall, the results of this scoping review are congruent with the literature at large and previous reviews touching upon the role of cats with regard to SARS-CoV-2. Other reviews also concluded that SARS-CoV-2 infection often leads to no recognizable symptoms at all or, if so, manifests as mostly mild disease with predominant respiratory signs [5,10,15,52,53,54]. Moreover, other reviews also support the finding of this study that, while cats are certainly susceptible to the virus and human-to-cat transmission regularly happens, there is currently no evidence of SARS-CoV-2 circulation among cats or from cats to humans [5,6,10,15,52,53,54].

This review only included studies that were found up until the 15 September 2021. Since then, a lot of literature has emerged. Thus, the databases were searched again on 7 February 2022 using the same search terms as before in an explorative search. The aim was to identify possible key studies that became available since September to discuss them in light of the findings of this review. However, while several studies have emerged since then that are congruent with the findings of this review, a scan of the literature did not reveal any radically new findings or key articles. Nevertheless, it narrows the scope of this review, that only studies published until September 2021 were included in this review. Moreover, the exclusion of studies that included other animal species such as dogs in addition to cats, poses another limitation to our review.

#### 4.2.1. Mink-to-Cat Transmission

One topic that was not addressed in the articles included in this review was mink-to-cat transmission of SARS-CoV-2; Some studies that investigated outbreaks of SARS-CoV-2 on mink farms also sampled cats in the surrounding areas to investigate whether the SARS-CoV-2-infected minks transmitted the virus to cats roaming in the area. However, these studies were not included in the literature review since multiple species, such as dogs, were tested. Van Aart et al. [55] took swabs of 89 feral cats that roamed in the proximity to different mink farms with a SARS-CoV-2 outbreak in the Netherlands. Three cats (3.4%) tested positive for SARS-CoV-2 using PCR testing. Viral sequencing of one swab sample confirmed that the sequence clustered with virus sequences obtained from minks on the farm. This strongly suggests that the cat contracted the virus from one of the minks as the feral cats were reported to not have had any contact with the humans living and working on the farms. As for the other two cats that tested positive, it is likely that infection was contracted from minks, but no definite conclusions can be drawn. Moreover, 11 of 62 tested feral cats were seropositive for SARS-CoV-2 antibodies, suggesting even more possible cases of mink-cat transmission—or cat-to-cat transmission for that matter. Also, Oreshkova et al. [56] found 7 of 24 stray cats in the proximity to two mink farms with SARS-CoV-2 outbreaks to be seropositive and one cat had a positive PCR test. However, it was not possible to generate a sequence from the sample taken, so there was no definite indication about the source of the virus. In contrast to these two studies, Boklund et al. [57] did not find any evidence of SARS-CoV-2 infection in the serum or via PCR tests in 30 stray cats in the surroundings of two mink farms in Denmark.

Taken together, the sporadic presence of SARS-CoV-2 antibodies in stray and feral cats roaming in the surroundings of different mink farms strongly indicate the possibility of mink-to-cat transmission.

#### 4.2.2. Seroprevalence of SARS-CoV-2 Antibodies among Cats

A key question that remains is how many cats get infected with SARS-CoV-2. Nine different studies investigated this topic by testing the prevalence of SARS-CoV-2 antibodies in different cat populations, which are summarized in Table 3. However, the results of the serosurveys are hard to generalize or compare to one another for multiple reasons. First, the studies took place in different areas of the world during different phases of the pandemic, thus leading to different epidemiological circumstances in the human population, consequently impacting the cat populations as well. Second, the study populations were vastly different, affecting the generalizability of the studies. Some studies looked at shelter cats, some at stray cats, some at pet cats, and some at pet cats with owners with a history of COVID-19. Third, the studies used different assays and tests to determine seropositivity for SARS-CoV-2, thus affecting the reported seroprevalence. Deng et al. [46], Jara et al. [25], Spada et al. [36], and Villanueva-Saz et al. [40] used different enzyme-linked immunosorbent assays (ELISAs), with considerable differences in methodology and reliability of the used assay. The four remaining studies also used different ELISAs, but confirmed positive ELISA results with a virus neutralization test (VNT), which is regarded as gold standard to reliably detect neutralizing antibodies [58].

Due to these vast differences between the serosurveys, the only reliable conclusion to be drawn from the studies is that overall, there was a relatively low prevalence of SARS-CoV-2 in the tested cats.

#### 4.2.3. The Role of Subclinical Infections

Most of the evidence regarding clinical manifestation of SARS-CoV-2 infection in cats was gathered by case studies. Of 20 case studies, all but 3 reported clinical symptoms. However, it must be considered, that in many case studies testing for SARS-CoV-2 in pet cats was performed in response to respiratory symptoms, thus possibly overestimating the prevalence of symptoms in infected cats. Further, the experimental studies did not report any respiratory signs in the infected cats and only Bao et al. [17] reported any kind of clinical symptoms in infected animals [14,23,45]. This gives further evidence for a possible underestimation of the number of subclinical infections by the case studies. The experimental studies also found that asymptomatic, infected cats were capable of transmitting SARS-CoV-2 to other cats, highlighting their potential role in transmission dynamics [14,17,24].

Overall, these results indicate that many SARS-CoV-2 infections in cats might be asymptomatic, which does not necessarily mean that they are not able to transmit the virus to other cats.

#### 4.2.4. Cat-to-Human Transmission

As mentioned before, none of the included studies found evidence of cat-to-human transmission. Neither have any other studies performed obtained such evidence. While this is somewhat reassuring as to the role of cats in the spread of SARS-CoV-2, it also raises some questions. How would one ever evidence a cat-to-human transmission of SARS-CoV-2? Due to ethical and study design considerations it is highly unlikely that evidence of such a zoonotic transmission event would ever be obtained. The most likely source of COVID-19 in humans is another infected human due to constant virus transmission among the human population and it would prove extremely challenging to evidence any other way. However, it is conceivable that transmission of the virus from cats to humans could occur via close contact, such petting, sharing food, or giving kisses or licks [59]. As noted by Totton, Sargeant and O’Connor [59], ethical reasons preclude the conduction of an experiment exposing an uninfected human to an infected cat. Therefore, cat-to-human transmission could only be demonstrated in real-life situations. Yet, real-life situations in which a cat could unequivocally be shown as source of SARS-CoV-2-infected are very limited at this point; As argued by Totton, Sargeant and O’Connor [59], an effective quarantine period followed by negative PCR and serological testing eliminating the possibility of an undetected infection in a person would be needed. Moreover, isolation of the person from all other sources of SARS-CoV-2 during the quarantine period, exposure to the cat, and up until diagnosis would be another prerequisite. Therefore, it is exceptionally doubtful that direct evidence of a zoonotic transmission event of SARS-CoV-2 between cats and human will ever be acquired.

#### 4.2.5. Cat-to-Cat Transmission

The biggest point of concern is the potential circulation of SARS-CoV-2 among cats. Even if cat-to-human transmission was possible, it would not be that concerning if SARS-CoV-2 is not circulating amongst cats. If SARS-CoV-2 was solely transmitted from humans to cats and the transmission chain did not go on, then there would be little possibility for the virus to spread and evolve. The emergence of a species-specific variant would be a lot less likely.

As shown by one case study and several experimental studies, cats can transmit the virus to other animals of their species [14,17,19,20,24]. However, the question whether cat-to-cat transmission happens on a regular basis in real-life situations, leading to circulation of the virus among the animals, is still not conclusively clarified. The serosurveys among colonies of stray cats and free-roaming, domestic cats conducted by Stranieri et al. [37] and Villanueva-Saz et al. [40] found a rather low seroprevalence compared to seroprevalence among humans in the same region of 0% and 3.5% respectively. Moreover, the serosurvey by van der Leij et al. [39] found a seroprevalence of only 0.8% among shelter cats even though they often live in close contact with other cats. These findings indicate that it is somewhat unlikely that there is sustained transmission of SARS-CoV-2 among cats. Further, the serial passaging experiment done in cats by Bao et al. [17] highlighted that while cats can transmit SARS-CoV-2 among several passages, the transmissibility and detected viral load weakened per passage. Moreover, antibody titers at 14 days post-infection were approximately eight times lower in first-passage recipient cats compared to inoculated cats. For cats in the second or later passage, antibody levels were even below detection. These findings further suggest that extended cat-to-cat transmission chains, as might for instance happen in feral cat colonies, are unlikely to be sustained.

Nevertheless, current data is not very extensive and the only point that is clearly evidenced, is the potential for cat-to-cat transmission. Thus, the potential of circulation of SARS-CoV-2 under natural conditions remains one of the biggest concerns. Specifically in light of an ever-evolving and changing virus, the potential circulation of SARS-CoV-2 among cats (in the future) cannot be ruled out.

#### 4.2.6. Evolution of SARS-CoV-2 in Cats

There is a constant race of arms between host and pathogen. Pathogens are under constant selective pressure to proliferate in the host environment, while host defense mechanisms constantly aim to prevent replication of the pathogen. Thus, evolution of SARS-CoV-2 is an unavoidable reality. The constant emergence of new variants of concern with the most recent example of the omicron variant illustrates this vividly [60]. Interspecies transmission specifically results in the rapid host adaptation of the virus, and continued transmission can accelerate mutation acquisition and novel strain emergence [61].

The study of Braun et al. [20] included in this review suggested the relative genetic stability of SARS-CoV-2 in cats. Nevertheless, the study showed some virus adaptations that are potentially adaptive to the cat host, such as the observed rapid fixation of Spike H655Y variant. This amino acid residue has previously been hypothesized to play a role in regulating Spike glycoprotein fusion efficiency, thus possibly aiding viral transmission [62,63,64]. Moreover, it has been shown to confer escape from different human SARS-CoV-2 antibodies in cell culture [62]. Notably, H655Y is also one of the key amino acid substitutions in the spike protein of the omicron variant and has been linked, amongst many other mutations, to higher transmissibility of the variant [65]. Another experimental study concerned with the genetic diversity arising and persisting of SARS-CoV-2 in different mammals found rapid adaptation and variant selection in cats in a serial passage experiment [66]. Interestingly, also in this study H655Y was detected in the viral RNA from three out of six cats.

Larger studies are warranted to investigate the genetic stability of SARS-CoV-2 and important evolutionary processes when transmitted among cats. Yet overall, it is clear that species- and context-specific adaptations of SARS-CoV-2 are likely to continue to emerge.

### 4.3. Zooming Out: The Wider Implications

Overall, cats take on a special position as one of the most popular pets worldwide that live in a close relation with humans. Among all animals that are regularly held as pets around the world, cats are the one most susceptible to SARS-CoV-2. However, SARS-CoV-2 is not (yet) optimally developed for the cat host. It does not seem like SARS-CoV-2 is widely circulating among cats at the moment, even though one ought to be careful to draw definite conclusions due the scarcity of data. Nevertheless, the big concern is that circulation of SARS-CoV-2 might establish in cats in the future, thus increasing the likelihood of the evolvement of new variants and transmission of those to humans. The prospect of cats as (silent) hosts for SARS-CoV-2 is truly concerning.

There is no evidence concerning the susceptibility of cats to the omicron variant yet. Possibly omicron, or potential future variants, might change infection dynamics among cats. As this variant of concern has been shown to transmit easier among humans, some researchers suspect that it might be easier transmittable in animal species as well [65,67]. However, this remains highly speculative at this point in time. A first step of investigation should be in vitro or in silico experiments regarding cat ACE2 receptor affinity to the omicron variant. Another step should be in vivo transmission experiments with this new SARS-CoV-2 variants among cats.

Interestingly, researchers have hypothesized that omicron might be a result of cross-species transmission due to its unusual constellation of changes in the virus, with more than 30 changes in the spike protein prominently responsible for entry into host cells [67,68,69]. However, also this hypothesis remains speculative. Considering that it is particularly hard to track the exact source of such virus strains, and there has been no success with any of the preceding variants of concern, it seems rather unlikely that this hypothesis will be substantiated in the future. Nonetheless, such concerns highlight the urgent need for widespread surveillance efforts of potentially susceptible animal populations.

While there have been no (known) cases of widespread circulation of SARS-CoV-2 among cats, or zoonotic transmission from cats to humans, there have been for other species. The most prominent example was the widespread transmission of the virus on several mink farms around the world and the zoonotic transmission of a mink-associated variant to humans in Denmark [4]. Fortunately, the mink-associated variant turned out to not show higher transmissibility, pathogenicity, or immune evasion capacities in comparison to other circulating strains among humans and did not spread very widely [70,71]. One could argue that the conditions provided by factory farming of minks are not comparable with the reality of how most cats live. Mink farms provide, in addition to a reasonably susceptible host, a very specific environment; The high number of animals in one place, the close contact between these animals, the inevitable contact to droppings and other bodily substances of animals housed in adjacent cages, and often immunocompromised animals provide optimal conditions for pathogen spread.

Outbreaks such as the ones in mink farms seem rather unlikely in cats as the situation of factory-farmed minks seems hardly comparable to the situation of cats. However, cats are bred on a large scale in many places around the world; Cats are often bred to be sold as pets, but also as for research, testing, and teaching [72,73,74]. Breeding facilities pose concerns regarding the spread of infectious diseases among cats due to close contact between many cats [75]. Certainly, there are vast differences in quality of breeding conditions and breeding locations where cats are bred and kept under cramped, unhygienic, and illness-promoting conditions are specifically concerning [74]. Monitoring of compliance with hygiene standards and increased surveillance for the circulation of SARS-CoV-2 in breeding facilities are of importance.

Conditions provided by factory farming or breeding locations favor the process of virus spread and evolution. However, they are no prerequisite for SARS-CoV-2 to establish in an animal population. Ultimately, any place where humans and susceptible animals have contact should generate concern and attention. This is strongly illustrated by multiple recent reports of widespread SARS-CoV-2 infections among white-tailed deer in 15 states of the US [51,76,77,78]. The studies indicate multiple, independent anthroponotic transmission events in different places, and continued deer-to-deer transmission of the virus. However, no zoonotic transmission from deer to humans has been evidenced. The circulation of SARS-CoV-2 among deer is the first wide-spread transmission among free-living animals that has been documented. Moreover, white-tailed deer on Staten Island of New York have been found to be infected with the omicron variant, making it the first reporting of the variant in wild animals [77]. White-tailed deer are widely distributed across the US and live near humans, giving rise to concerns regarding the new pathways this host opens for viral adaptation and evolution and the dangers of zoonotic transmission of the virus back to humans.

In light of the potential danger of virus circulation in animal populations for humans, effective preventive measures are urgently needed. When looking at the example of SARS-CoV-2 outbreaks on mink-farms, Denmark and Netherland for instance resorted to the “solution” of wide-spread culling of the animals [79,80]. The Netherlands even introduced a complete ban on mink farming [81]. Arguably, mass culling is not a sustainable solution for the problem at hand, but is, if anything, a quick fix to symptoms of a much more systematic problem [81]. Humanity can hardly prevent the circulation of SARS-CoV-2 in animals and resulting animal-human transmission simply through measures such as widespread culling. Moreover, such a short-term solution would not only impose serious ethical and ecological concerns, but also meet large resistance in the human population when it comes to cats or other animals that lie close to the heart of many people. Given the fondness many people have for cats, the culling of feral and stray cats, but specifically of domestic cats, can be expected to meet large resistance in the general population [82].

There is a need for change of paradigm when it comes to the defense strategy against zoonotic diseases such as SARS-CoV-2 at the human-animal interface. Currently, a large focus is on determining and eliminating animal culprits, for instance through widespread culling of animal populations in question. However, this focus is ineffective. Despite large efforts, the zoonotic origin of SARS-CoV-2 remains elusive [83]. No SARS-CoV-2 circulation in animals that could have acted as intermediate species to transmit the virus to humans has been found. Blaming animals for zoonotic virus emergence and adaptation may result in useless, if not highly damaging, culling, mass slaughter and even loss of biodiversity [84]. The focus should be on the role that humans play in the spread and transmission of the virus to animals [83,84]. Certainly, monitoring transmission dynamics in animals is highly necessary. However, the focus should be on the more relevant, human-related factors; The real triggers that drive zoonotic virus emergence and subsequent epidemics/pandemics are society-driven human-animal contacts in combination with amplifying factors provided by modern human societies, e.g., land conversion, factory farming of animals, international mobility, markets, or international trade [84]. There is the urgent need for an integrated, systemic approach to SARS-CoV-2. A One Health approach, which recognizes the interdependence of the health of humans, animals, and the environment is needed to understand and prevent risks that originate at the interfaces between humans, cats, and the environment [85,86].

There is no quick fix to the concern cats, and other animals, pose with regard to SARS-CoV-2. However, there are an array of measures that can be introduced. Firstly, there is an urgent need for more comprehensive surveillance of cats and other susceptible animal populations to identify ecological transmission networks [87]. It is important to ensure detection of circulation of virus among animals and transmission events. For cats in specific this would mean monitoring of stray and feral cat populations for SARS-CoV-2. Moreover, sporadic sampling of pet cats is needed. Specifically, cats from owners with confirmed COVID-19 infection should be sampled more often, including asymptomatic animals. Studies such as the serosurvey by Jara et al. [25] indicate, that a substantial amount of pet cats of SARS-CoV-2-positive owners become infected. To avoid circulation of SARS-CoV among cats, infection of domestic cats should be avoided as much as possible. Thus, in addition to wide-spread surveillance efforts, authorities should clearly advise citizens to avoid close contact to their cats in the event of a suspected of confirmed case of COVID-19, as advised by the European Advisory Board on Cat Disease [88]. More awareness among the population would be helpful in this regard, but it is a fine line to walk to not stir excessive fear and avoid potential negative consequences such as abandonment of domestic cats. Another option, which is currently explored, is vaccination development for domestic and captive animals to prevent the introduction into feral populations or other wildlife [89]. However, all these measures will most likely not be enough to completely prevent circulation of SARS-CoV-2 in animals and potential animal-human transmission. To truly make a difference, contact with animals needs to be minimized and high-risk practices such as factory farming need to be avoided.

## 5. Conclusions

In conclusion, cats seemingly play a very limited role in the spread of SARS-CoV-2. While cats are susceptible to the virus and reverse zoonotic transmission events from humans to cats happen regularly, there is currently no evidence of SARS-CoV-2 circulation among cats. Nevertheless, close surveillance is warranted specifically due to the unpredictable nature of virus development and the consequent influence of transmission dynamics concerning the cat hosts. More immediately, current knowledge implicates precautionary measures of cat owners regarding the handling of domestic cats when diagnosed or suspected of having COVID-19.

In the future, more research about the role of cats with regard to SARS-CoV-2 is urgently needed to close knowledge gaps. Specifically, large-scale studies that include a greater number of cats are necessary to solidify current evidence gathered from individual case studies. Another important gap that needs to be tackled is the underrepresentation of stray, feral, and shelter cats among the studies to shine more light on the potential roles these cats play in SARS-CoV-2 transmission dynamics. Widespread studies in these cat populations are specifically helpful to investigate the important topic of natural cat-to-cat transmission dynamics as there is currently little knowledge regarding this topic. Beyond more research, surveillance, and careful interaction with pet cats, we need to tackle the actual root problem more. We need to carefully examine and rethink the current relationship humanity has with animals and ecosystems at large.

## Figures and Tables

**Figure 1 animals-12-01413-f001:**
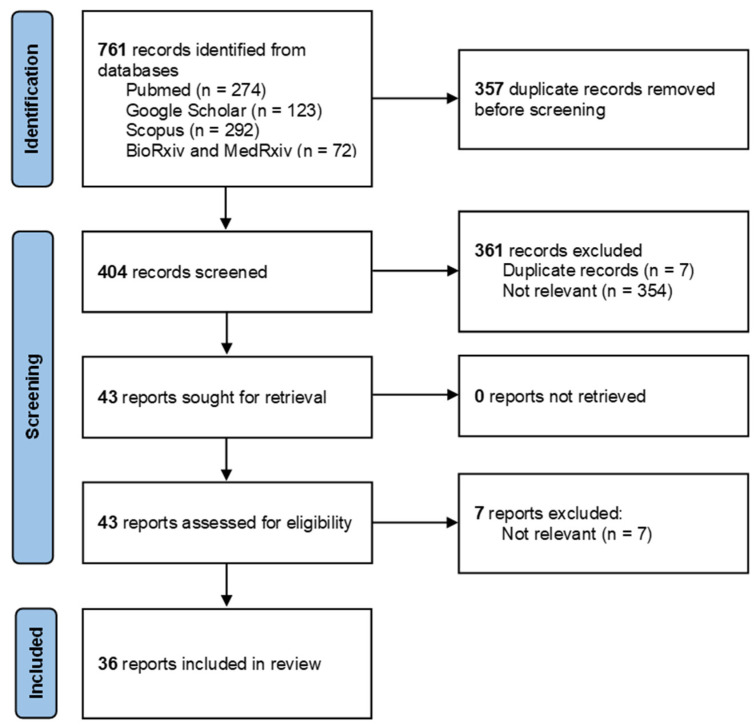
PRIMA flow diagram of article selection process.

**Figure 2 animals-12-01413-f002:**
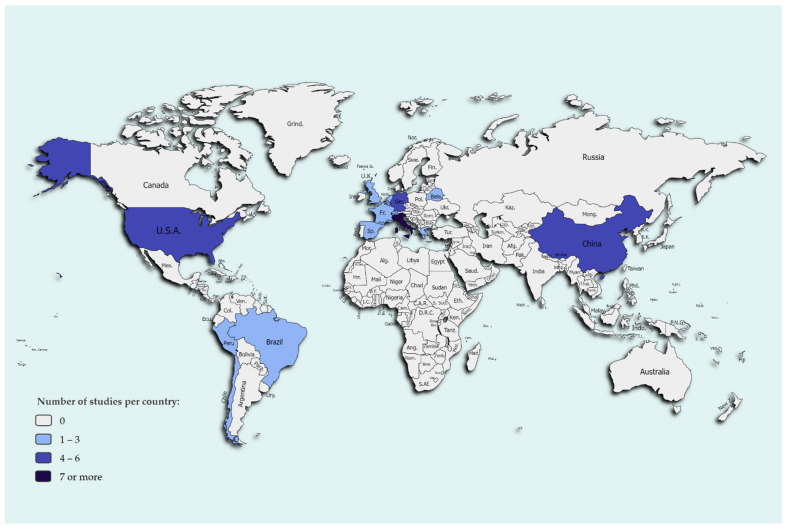
Number of studies included in the review per country. With seven studies, the highest number of studies was performed in Italy. In both China and Germany, there were five studies, while in the USA, four studies were conducted. In all other countries (Belarus, Belgium, Brazil, Chile, Greece, Netherlands, Northern Cyprus, Peru, Spain, Switzerland, and the United Kingdom) either one or two studies were performed.

**Table 1 animals-12-01413-t001:** Inclusion and exclusion criteria for the scoping review.

Topic	Inclusion Criteria	Exclusion Criteria
Language	English	Language other than English
Publication date	2019–2021	Before 2019
Study design	All study designs	
Research type	Primary research	Secondary research
Topic area	Role of cats and their interaction with humans and other animals with regard to the spread of SARS-CoV-2	Assessment of cats as animal model for human SARS-CoV-2 infection Articles dealing solely with human well-being derived from contact to pet cats Development/evaluation of biochemical assays for SARS-CoV-2 In vitro/in silico biochemical analysis relating to the cat ACE2 receptor
Study participants	Cats	Other animals than cats Multiple animal species
Publication type	Published and peer-reviewed articles Research letters Pre-prints	Preprints that have later been published as articles in peer-reviewed journals

**Table 2 animals-12-01413-t002:** Overview of eligible records and their aims.

First Author	Year	Location	Study Aim	Study Type	N *	Reference
**Peer-reviewed articles**						
Bao	2021	China (Beijing)	Evaluation of the cat-to-cat transmissibility of SARS-CoV-2 upon serial passaging	Case Study	18	[17]
Bessière	2021	France (Toulouse)	Enhancing understanding of cats’ role in COVID-19 epidemiology through monitoring of viral shedding in five different cats and their quarantined, COVID-19 positive owners	Case Study	5	[18]
Brandão	2021	Brazil (Alta Floresta city)	Reporting SARS-CoV-2 detection using PCR testing and clinical manifestations in two domestic cats	Case Study	2	[19]
Braun	2021	USA (Wisconsin)	Characterization of viral genetic diversity arising, persisting, and being transmitted in domestic cats to model the evolutionary capacity of SARS-CoV-2 within and between hosts	Experimental	6	[20]
Chaintoutis	2021	Greece (Thessaloniki)	Reporting the first case of SARS-CoV-2 infection in a cat in Latin America	Case Study	3	[21]
Curukoglu	2021	Northern Cyprus	Investigation of the course of SARS-CoV-2 infection in naturally exposed cats in small household setting	Case Study	1	[22]
Gaudreault	2020	USA (Manhattan, Kansas)	In-depth study of SARS-CoV-2 infection, associated disease and transmission in domestic cats	Experimental	10	[23]
Gaudreault	2021	USA (Manhattan, Kansas)	Investigation of possibility of SARS-CoV-2 re-infection in previously infected cats	Experimental	11	[24]
Hosie	2021	United Kingdom	Finding evidence of SARS-CoV-2 infection in cats from the UK	Case Study	2	[11]
Jara	2021	Peru (Lima)	Demonstrating presence of neutralizing antibodies against SARS-CoV-2 in cats whose owners have been infected with SARS-CoV-2	Cross-sectional	41	[25]
Keller	2021	Germany	Reporting of first case of a cat infected with SARS-CoV-2 alpha variant of concern	Case Study	1	[26]
Klaus	2021	Italy	Investigation of the case of a cat with SARS-CoV-2 with underlying B-cell lymphoma	Case Study	1	[27]
Klaus	2021	Switzerland (Zurich)	Reporting of first SARS-CoV-2 infections in cats in a COVID-19 affected household in Switzerland	Case Study	2	[28]
Michelitsch	2020	Germany	Assessing the incidence of naturally occurring human-to-cat transmission of SARS-CoV-2 through serological testing	Cross-sectional	920	[29]
Michelitsch	2021	Germany	Monitoring the occurrence of SARS-CoV-2 transmission between humans and cats during second pandemic wave	Cross-sectional	1173	[30]
Musso	2020	Italy	Reporting the first natural infection of a cat with SARS-CoV-2 in Italy	Case Study	1	[31]
Natale	2021	Italy	Reporting and describing of a case of symptomatic, natural SARS-CoV-2 infection in a cat	Case Study	1	[32]
Neira	2021	Chile (Santiago City)	Reporting of case of three cats in a household with SARS-CoV-2 infection	Case Study	3	[33]
Pagani	2021	Italy	Description of case of human-to-cat SARS-CoV-2 transmission and full-genome analysis of virus from human and cat	Case Study	1	[34]
Segalés	2020	Spain	Description of case of a cat with severe respiratory symptoms and thrombocytopenia in a SARS-CoV-2-infected household	Case Study	2	[35]
Spada	2021	Italy (Lombardy)	Investigation of SARS-CoV-2 infection in a stray cat population through serological survey and PCR testing	Cross-sectional	99	[36]
Stranieri	2021	Ital (Lodi province)	Evaluation of the presence of SARS-CoV-2 RNA and antibodies against SARS-CoV-2 in free roaming cats belonging to colonies in an area highly affected by the COVID-19 pandemic	Cross-sectional	99	[37]
Subotsina	2021	Belarus	Determination of clinical, pathoanatomical, and histological features in domestic cats with SARS-CoV-2 infection	Case Study	15	[38]
Van der Leij	2021	Netherlands	Determination of seroprevalence of SARS-CoV-2 in Dutch shelter cats	Cross-sectional	240	[39]
Villanueva-Saz	2021	Spain (Zaragoza)	Furthering the knowledge about the role played by stray cats in the context of SARS-CoV-2 and possible predisposing of cats to SARS-CoV-2 infection through concomitant infections	Cross-sectional	114	[40]
Zhang	2020	China (Wuhan)	Investigation of SARS-CoV-2 infection in cats during COVID-19 outbreak in Wuhan using serological detection methods	Cross-sectional	141	[41]
Zoccola	2021	Italy	Description of Italian human-to-cat transmission of SARS-CoV-2 alpha variant of concern	Case Study	1	[42]
**Scientific communication items**						
Barrs	2020	China (Hong-Kong)	Reporting PCR testing results for cats from COVID-19 households	Cross-sectional	50	[43]
Carlos	2021	Brazil	Reporting the first case of SARS-CoV-2 infection in a cat in Latin America	Case Study	1	[44]
Chiba	2021	USA (Wisconsin)	Further characterization of the biology of SARS-CoV-2 in cats through experimental inoculation of cats with the virus and subsequent histopathological examination and re-infection experiment	Experimental	17	[45]
Deng	2020	China	Elucidating the role of domestic cats in SARS-CoV-2 transmission in China through serological survey	Cross-sectional	423	[46]
Gargliany	2020	Belgium	Investigation of SARS-CoV-2 infection and illness in a domestic cat in Belgium	Case Study	1	[47]
Halfmann	2020	Germany (Hamburg)	Evaluation of nasal shedding of SARS-CoV-2 from inoculated cats and subsequent transmission by direct contact to naïve cats	Experimental	6	[14]
Schulz	2021	Germany	Reporting the case of three domestic cats in a retirement home and likely natural human-to-cat SARS-CoV-2 transmission	Case Study	3	[48]
**Unpublished pre-prints**						
Akhmetzhanov	2020	China (Wuhan)	Resolving some uncertainty around SARS-CoV-2 transmission potential to and between cats and quantification of transmission strength.	Case Study	143	[49]
Fritz	2021	France	Investigation of clinical and biological features of SARS-CoV-2 infection in two mildly symptomatic cats from SARS-CoV-2 household	Case Study	2	[50]

* N = number of cats included in study.

**Table 3 animals-12-01413-t003:** Cross-sectional studies investigating seroprevalence of SARS-CoV-2 antibodies.

First Author	Year	Seropre-Valence	N Positive Samples (N Total Samples) *	Method **	Type of Cat	Timeframe	Study Place	Reference
Deng	2020	0%	0 (423)	ELISA	Domestic	February 2020–April 2020	China	[46]
Jara	2021	31.7%	13 (41)	ELISA	Domestic with owners with COVID-19 history	August 2020–April 2021	Lima, Peru	[25]
Michelitsch	2020	0.7%	6 (920)	ELISA + VNT	Domestic	April 2020–September 2020	Germany	[29]
Michelitsch	2021	1.4%	16 (1173)	ELISA + VNT	Domestic	September 2020–February 2021	Germany	[30]
Spada	2021	1.0%	1 (99)	ELISA	Stray, shelter	December 2019–February 2021	Italy (Lombardy)	[36]
Stranieri	2021	0%	0 (99)	ELISA + VNT	Stray, free-roaming domestic	December 2019–December 2020	Italy (Lodi province)	[37]
Van der Leij	2021	0.8%	2 (240)	ELISA + VNT	Shelter	August 2020–February 2021	Netherlands	[39]
Villanueva-Saz	2021	3.5%	4 (114)	ELISA	Stray	January 2020–October 2020	Zaragoza, Spain	[40]
Zhang	2020	10.8%	11 (102)	ELISA + VNT	Shelter, domestic, domestic with owners with COVID-19 history	January 2020–March 2020	Wuhan, China	[41]

* N = Number; ** The column “Method” indicates the kind of test or assay used to determine seropositivity. “ELISA + VNT” means that all samples were tested using ELISA and ELISA-positive samples were confirmed using VNT.

## Data Availability

Not applicable.

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
