# Peer review of "Cats and SARS-CoV-2: A Scoping Review"

_animals, 2022, doi:10.3390/ani12111413_

Round 1

Reviewer 1 Report

The manuscript entitled “Review Cat and SARS-CoV-2 a scoping review” provide a rather complete and very well-written review of the literature. Several review on the topics are already available, but the qulatity of this review justify its publication. The aim of the review is to “provide a comprehensive overview about what is known concerning the role of cats regarding the spread of SARS-CoV-2 to guide further research and inform policy makers”. Overall, this is an excellent review, consistent with its objectives, with a very good description of the methodology exposing the reasons for one publication to be included in the scope of the manuscript. Also, the authors provided the readers with tables summarizing the publications described in the review, with the general characteristics of the publications, which is very useful.

I regret that most of the articles cited in this review were available prior September 2021, and I am afraid that this would be somewhat outdated when this review comes out. (example: line 210 : “None of the included studies reported the detection of any other variant of concern.” Which is not the case anymore (Krafft et al., viruses, 2021 and other).

However, in §554-559, authors mentioned that they scanned the literature in february 2022, and noticed no major change in the newly available publications.

I also kind of disagree to remove the studies not focusing exclusively on cat as most of the published works consider pets as a whole (cats and dogs and sometimes ferrets), especially the work done during the first months of the pandemic. This may lead to important gaps in this review, especially in terms of seroprevalence evaluation, considering that most studies included cats and dogs. I think that adding a selection of the most important findings from this sort of study would increase the visibility of this review.

The following references (published prior september 2021) are not included in the review, some of them may have been discarded due to the criteria chosen by the authors (especially removing publications with different species?). However, some of the following publications gave important princeps results. Maybe there are objectives reasons for dismissing them ?

High prevalence of SARS-CoV-2 antibodies in pets from COVID-19+ households.

Fritz M, Rosolen B, Krafft E, Becquart P, Elguero E, Vratskikh O, Denolly S, Boson B, Vanhomwegen J, Gouilh MA, Kodjo A, Chirouze C, Rosolen SG, Legros V, Leroy EM.

One Health. 2021 Jun;11:100192. doi: 10.1016/j.onehlt.2020.100192. Epub 2020 Nov 4.

PMID: 33169106

One Health Investigation of SARS-CoV-2 Infection and Seropositivity among Pets in Households with Confirmed Human COVID-19 Cases-Utah and Wisconsin, 2020.

Goryoka GW, Cossaboom CM, Gharpure R, Dawson P, Tansey C, Rossow J, Mrotz V, Rooney J, Torchetti M, Loiacono CM, Killian ML, Jenkins-Moore M, Lim A, Poulsen K, Christensen D, Sweet E, Peterson D, Sangster AL, Young EL, Oakeson KF, Taylor D, Price A, Kiphibane T, Klos R, Konkle D, Bhattacharyya S, Dasu T, Chu VT, Lewis NM, Queen K, Zhang J, Uehara A, Dietrich EA, Tong S, Kirking HL, Doty JB, Murrell LS, Spengler JR, Straily A, Wallace R, Barton Behravesh C.

Viruses. 2021 Sep 12;13(9):1813. doi: 10.3390/v13091813.

PMID: 34578394 Free PMC article.

Report of One-Year Prospective Surveillance of SARS-CoV-2 in Dogs and Cats in France with Various Exposure Risks: Confirmation of a Low Prevalence of Shedding, Detection and Complete Sequencing of an Alpha Variant in a Cat.

Krafft E, Denolly S, Boson B, Angelloz-Pessey S, Levaltier S, Nesi N, Corbet S, Leterrier B, Fritz M, Leroy EM, Gouilh MA, Cosset FL, Kodjo A, Legros V.

Viruses. 2021 Sep 3;13(9):1759. doi: 10.3390/v13091759.

PMID: 34578341 Free PMC article.

Long-term persistence of neutralizing SARS-CoV-2 antibodies in pets.

Decaro N, Grassi A, Lorusso E, Patterson EI, Lorusso A, Desario C, Anderson ER, Vasinioti V, Wastika CE, Hughes GL, Valleriani F, Colitti B, Ricci D, Buonavoglia D, Rosati S, Cavaliere N, Paltrinieri S, Lauzi S, Elia G, Buonavoglia C.

Transbound Emerg Dis. 2021 Sep 1:10.1111/tbed.14308. doi: 10.1111/tbed.14308. Online ahead of print.

PMID: 34469620 Free PMC article.

First detection and molecular analysis of SARS-CoV-2 from a naturally infected cat from Argentina.

Fuentealba NA, Moré G, Bravi ME, Unzaga JM, De Felice L, Salina M, Viegas M, Nabaes Jodar MS, Valinotto LE, Rivero FD, Di Lullo D, Pecoraro M, Panei CJ.

Vet Microbiol. 2021 Sep;260:109179. doi: 10.1016/j.vetmic.2021.109179. Epub 2021 Jul 8.

PMID: 34271305 Free PMC article.

Severe SARS-CoV-2 Infection in a Cat with Hypertrophic Cardiomyopathy.

Carvallo FR, Martins M, Joshi LR, Caserta LC, Mitchell PK, Cecere T, Hancock S, Goodrich EL, Murphy J, Diel DG.

Viruses. 2021 Jul 31;13(8):1510. doi: 10.3390/v13081510.

PMID: 34452375 Free PMC article.

SARS-CoV-2 Infection in Dogs and Cats from Southern Germany and Northern Italy during the First Wave of the COVID-19 Pandemic.

Klaus J, Zini E, Hartmann K, Egberink H, Kipar A, Bergmann M, Palizzotto C, Zhao S, Rossi F, Franco V, Porporato F, Hofmann-Lehmann R, Meli ML.

Viruses. 2021 Jul 26;13(8):1453. doi: 10.3390/v13081453.

PMID: 34452319 Free PMC article.

Cross-Sectional Serosurvey of Companion Animals Housed with SARS-CoV-2-Infected Owners, Italy.

Colitti B, Bertolotti L, Mannelli A, Ferrara G, Vercelli A, Grassi A, Trentin C, Paltrinieri S, Nogarol C, Decaro N, Brocchi E, Rosati S.

Emerg Infect Dis. 2021 Jul;27(7):1919-1922. doi: 10.3201/eid2707.203314. Epub 2021 May 11.

PMID: 33974535 Free PMC article.

Seroprevalence of SARS-CoV-2 infection among pet animals in Croatia and potential public health impact.

Stevanovic V, Vilibic-Cavlek T, Tabain I, Benvin I, Kovac S, Hruskar Z, Mauric M, Milasincic L, Antolasic L, Skrinjaric A, Staresina V, Barbic L.

Transbound Emerg Dis. 2021 Jul;68(4):1767-1773. doi: 10.1111/tbed.13924. Epub 2020 Nov 28.

PMID: 33191649 Free PMC article.

Absence of SARS-CoV-2 RNA and anti-SARS-CoV-2 antibodies in stray cats.

Stranieri A, Lauzi S, Giordano A, Galimberti L, Ratti G, Decaro N, Brioschi F, Lelli D, Gabba S, Amarachi NL, Lorusso E, Moreno A, Trogu T, Paltrinieri S.

Transbound Emerg Dis. 2021 Jun 25:10.1111/tbed.14200. doi: 10.1111/tbed.14200. Online ahead of print.

PMID: 34170624 Free PMC article.

SARS-CoV-2 infection in cats and dogs in infected mink farms.

van Aart AE, Velkers FC, Fischer EAJ, Broens EM, Egberink H, Zhao S, Engelsma M, Hakze-van der Honing RW, Harders F, de Rooij MMT, Radstake C, Meijer PA, Oude Munnink BB, de Rond J, Sikkema RS, van der Spek AN, Spierenburg M, Wolters WJ, Molenaar RJ, Koopmans MPG, van der Poel WHM, Stegeman A, Smit LAM.

Transbound Emerg Dis. 2021 Jun 3:10.1111/tbed.14173. doi: 10.1111/tbed.14173. Online ahead of print.

PMID: 34080762 Free PMC article.

SARS-CoV-2 Infections and Viral Isolations among Serially Tested Cats and Dogs in Households with Infected Owners in Texas, USA.

Hamer SA, Pauvolid-Corrêa A, Zecca IB, Davila E, Auckland LD, Roundy CM, Tang W, Torchetti MK, Killian ML, Jenkins-Moore M, Mozingo K, Akpalu Y, Ghai RR, Spengler JR, Barton Behravesh C, Fischer RSB, Hamer GL.

Viruses. 2021 May 19;13(5):938. doi: 10.3390/v13050938.

PMID: 34069453 Free PMC article.

Serologic Screening of Severe Acute Respiratory Syndrome Coronavirus 2 Infection in Cats and Dogs during First Coronavirus Disease Wave, the Netherlands.

Zhao S, Schuurman N, Li W, Wang C, Smit LAM, Broens EM, Wagenaar JA, van Kuppeveld FJM, Bosch BJ, Egberink H.

Emerg Infect Dis. 2021 May;27(5):1362-1370. doi: 10.3201/eid2705.204055.

PMID: 33900184 Free PMC article.

Frequency of respiratory pathogens and SARS-CoV-2 in canine and feline samples submitted for respiratory testing in early 2020.

Michael HT, Waterhouse T, Estrada M, Seguin MA.

J Small Anim Pract. 2021 May;62(5):336-342. doi: 10.1111/jsap.13300. Epub 2021 Jan 31.

PMID: 33521974 Free PMC article.

Investigation of SARS-CoV-2 infection in dogs and cats of humans diagnosed with COVID-19 in Rio de Janeiro, Brazil.

Calvet GA, Pereira SA, Ogrzewalska M, Pauvolid-Corrêa A, Resende PC, Tassinari WS, Costa AP, Keidel LO, da Rocha ASB, da Silva MFB, Dos Santos SA, Lima ABM, de Moraes ICV, Mendes Junior AAV, Souza TDC, Martins EB, Ornellas RO, Corrêa ML, Antonio IMDS, Guaraldo L, Motta FDC, Brasil P, Siqueira MM, Gremião IDF, Menezes RC.

PLoS One. 2021 Apr 28;16(4):e0250853. doi: 10.1371/journal.pone.0250853. eCollection 2021.

PMID: 33909706 Free PMC article.

In addition to these important comments, I have a few issues listed below that can be easily fixed.

Simple summary:

Use “SARS-CoV-2” instead of “novel coronavirus”

Line 19 : Maybe replace “experiments” by “experimental infection”

Abstract

Maybe add the objectives of the review in the abstract?

Table 1

Top of second page cannot be read (multiple lines overlap)

Starting line 172 and further: check title numbering

Line 227 is more about the seroprevalence evaluation than SARS-CoV-2 induced immunity

Line 257: “Moreover, the viral load was insufficient to lead to transmission to co-housed cats“. I would like the authors to elaborate on the evidence supporting this statement (I don’t think that a minimal viral load has been defined to allow cat-to-cat transmission, other reasons can explain the absence of infection.

Line 273 “The cat Musso, et al. [31] reported about eventually died due to the severe clinical manifestations“clarify the sentence

Line 283: Maybe “possibly” can be removed because it is know well known that most cats are asymptomatic.

Line 338 and 360: While the rest of the titles is factual, the choice of the titles 4.4 and 4.5 are not very clear (ex : “The relation between humans and cats with regard to SARS-CoV-2” is quite hard to get the subject of the paragraph from the tittle only)

Line 374 change “PCR” by “RT-qPCR” or equivalent

Line 561 : “5.3.1 Changed role of cats as companion animals during the COVID-19 pandemic”I don’t think this paragraph really fits in the scope of the review

Since the authors conducted most of the review based on publications searched before sept 2021, but also added some papers after this date, it would be easier for the reader to have a date instead of “up until now” (ex: line 609, line 648)

Line 1098: Format ref 81

Reviewer 2 Report

Authors aims to determine the role of cats with regard to the spread of SARS-CoV-2, from a wide perspective.

Specific minor comments/suggestions are highlighted in the attached .pdf file.

The possibility of reviewing this work is appreciated, considering its level and the information provided. This is a very interesting topic, considering what has been described about felines in the context of the COVID-19 pandemic.

It is important to advance in knowledge generation about the role of animals in the current pandemic and how these relationships should be addressed in the face of future pandemics of zoonotic origin.
